# Effect of Operating Parameters on the Mulching Device Wear Behavior of a Ridging and Mulching Machine

Qinxue Zhao, Fei Dai *, Ruijie Shi, Wuyun Zhao, Pengqing Xu, Huan Deng and Haifu Pan

College of Mechanical and Electrical Engineering, Gansu Agricultural University, Lanzhou 730070, China; zhaoqx@st.gsau.edu.cn (Q.Z.); shirj@gsau.edu.cn (R.S.); zhaowy@gsau.edu.cn (W.Z.); 15214126363@163.com (P.X.); 17797691659@163.com (H.D.); 18198029608@163.com (H.P.)
* Correspondence: daifei@gsau.edu.cn; Tel.: +86-9317677809

**Abstract:** To conduct an in-depth investigation of the impact of various operating parameters on mulching device wear during the operation of full-film dual-row ridging and mulching machine mulching, this paper employed EDEM software to create a 3D discrete element model of how a mulching device interacts with the soil on the seed bed and simulated the dynamic process of the interaction between the mulching device and the soil during the mulching operation. We analyzed the cladding wear process between the cladding device and the cladding sand particles, and two areas of impact wear on the overburden conveyor housing and areas of wear on the chute deflector scratches were detected. A three-factor, three-level Box–Behnken experimental design approach was used, with mathematical modeling of the relationship between the scraper conveyor lifting line speed, seed bed cover, scraper spacing, and wear of the cover device, finding the optimal combination of operating parameters for mulching devices. The results of the simulation test indicated that the mulching device experienced a minimum wear of $0.958 \times 10^{-3}$ mm at a lifting line speed of $0.7$ m·s$^{-1}$ for the scraper conveyor, a mulching volume of $2.55$ kg·s$^{-1}$ for the seed bed, and a scraper spacing of 98 mm. The results of the field trial validation showed that, in a comparison between simulated wear parts and a mulching operation prototype of the same two wear parts, the established discrete element model appeared reasonable concerning the structural parameters, with a feasible abrasion mechanism process of sand particles on the soil-covering devices, demonstrating the model's reliability and validity. It can serve as a guide for optimizing the design of mechanized full-film dual-furrow seed bed mulching operation.

**Keywords:** full-film duopoly; discrete element method; wear; mulching device; mulching process; parameter optimization

## 1. Introduction

Dryland full-film double-monopoly furrow cultivation technology is a novel method developed from traditional film cover planting [1]. The technique involves rotary tillage, furrowing, full-film cover, mulch suppression, and fertilization and seeding in the furrow under the film, to ensure optimal precipitation utilization by the crop [2]. The technology enhances seed bed construction, with film evaporation suppression, with the ability to collect flows on the ridge surface and temperature accumulation underneath the film, this technique achieves effective water storage and moisture conservation. It also effectively resists drought by collecting rain, increases the seed bed's temperature, and improves the fertilizer and water utilization rate [3–5]. Since 2003, full-membrane duopoly cultivation technology for dryland farming has rapidly gained popularity at a large scale in Northwest China [6]. As such, the area designated for demonstrating and popularizing full-film ridge cultivation technology in Gansu Province alone is expected to extend to approximately 15 million mu in 2023. In recent years, the dry zone of Northwest China's expanding region has seen an increase in the diversity of ridging and mulching machines due to the implementation of full-film double-monopoly furrow technology and its support in seed bed construction [7].

A mulching device is a crucial part of ridge mulching equipment. This device is used in the construction process of mechanized double-row furrow seed beds to carry out longitudinal laying of mulching soil belts, transverse girdle mulching, and other operations. Its performance plays a significant role in realizing full-film double-row furrow planting bed mechanization, which is essential for effective operation [8]. Wear and tear are the primary cause of agricultural machinery component failures, accounting for around 80% of such failures [9]. During the operation of a full-film double-row furrow mulching combine, owing to the intricate and fluctuating nature of the operational settings, coupled with the existence of sand and stones within the soil particles [10] and under the combined effect of continuous impact and sliding friction of the soil on the mulcher, localized severe wear can occur, with the failure of the mulching device and the decreased effectiveness of the mulching, as well as constraints on the quality of ridging and subsequent seeding operations, which, in turn, impacts crop growth and yields [11].

In recent years, the discrete element method has been widely applied to analyze and study the wear problem between particles and components. Chen Zuxiang and Wang Xuewen [12] examined the wear distribution pattern of the middle groove in a scraper conveyor using EDEM software, this provided a reference for the structural design of future scraper conveyors. Yanqiang Zhang [13] took the bucket teeth of the WK-75 mining excavator as a research object and analyzed the most wearable parts and the wear pattern of the bucket teeth using EDEM. Cleary et al. [14] found that the wear behavior of HICOM® mills was influenced by the liner design using the discrete cell method. Zhang Chuangchuang et al. [15] used a discrete element approach to investigate the wear characteristics of a ploughshare and obtained the conclusion that the effect of ploughing speed on the wear of the share plough was greater than the effect of soil density on the wear of the share plough. It is now customary to examine the wear of various structures using the discrete element method. However, there is scarce research available on the wear loss efficacy of mulching devices in the dryland full-film duopoly furrow planting mode. As a result, few research findings can be used as references [16]. Therefore, investigating the wear mechanism of mulching devices is crucial in optimizing the performance of a full-film dual-monopoly mulching combine and enhancing its operational efficiency.

In this paper, combining discrete element simulation methods with actual field trials, a precise meta-model is developed to examine the interaction between the mulching machine's mulching device and the soil of a seed bed, based on the outcomes of preceding research, with simulation of the mulching process of a mulching device, predicting the areas of the mulching unit's delivery casing and chute that are prone to wear. Mathematical modeling was implemented to understand the correlation between the scraper conveyor lifting line speed, seed bed cover, scraper spacing, and wear on different components of the mulching device, including the chute and transfer shell. The study utilized factorial level tests and the response surface method to derive results, seeking the optimal combination of operating parameters for mulching operations of a mulching machine. The simulation results' validity was confirmed through field experiments, which also analyzed and explored how various operating parameters affected the mulching device's wear loss efficiency, where this analysis aims to provide a reference for enhancing the design and optimizing the performance of the mulching device in full-film double-monopoly furrow mulching machines.

## 2. Mulching Machine Structure and Mulching Working Principle

A full-film double-monopoly furrow mulching combine machine with horizontal belt mulching includes a suspension device, a fertilizer discharge system, a spraying system, furrow and horizontal belt mulching device, a ridge mulching device, rotary tillage cutter set, soil up shovel, scraper conveyor, ground wheel, furrow suppression perforating wheel, film side suppression wheel, and other components [17]. The overall structure of the combine is shown in Figure 1a, and the structural components of the mulching device are shown in Figure 1b.

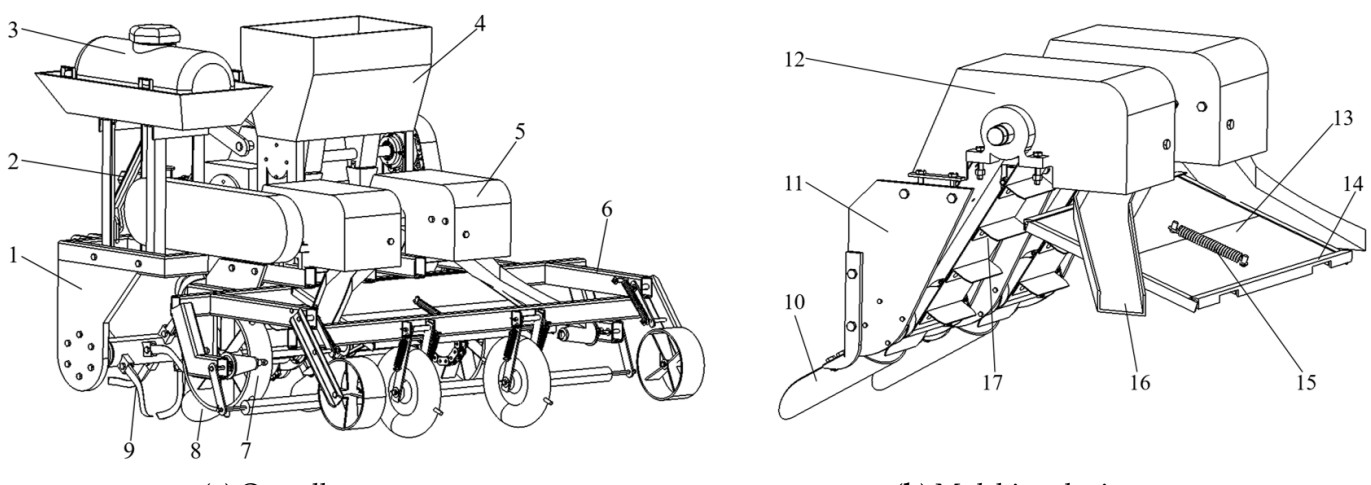

(**a**) Overall structure                    (**b**) Mulching devices

**Figure 1.** Structural diagram of a ridging and mulching machine and mulching device: 1. rack, 2. suspension device, 3. spraying device, 4. fertilizer applicator, 5. mulching device, 6. suppression device, 7. floor wheel, 8. ridge-raising device, 9. rotary plowing devices, 10. Shovels, 11. side panels, 12. earth transfer shell, 13. mulchers for ridges and cross girdles, 14. center-mounted chute, 15. reset spring, 16. side-mounted chute, 17. scraper conveyor.

The mulching operation is a critical part of the process of constructing a seed bed in a duopoly furrow. The mulching machine operation utilizes a four-wheeled tractor with rear suspension mode, and the tractor power is transmitted to the rotary tillage cutter shaft, which then drives the rotary tillage cutter group to loosen the seed bed soil. At the same time, the shovel for removing soil advances with the entire machine to make the exposed furrow in the loosened seedbed. The mulching process is completed through the synchronized rotation of the film hanging device, which is uniformly laid on the small ridge's body. This film extends to the middle of the two sides of the large ridge, ensuring uniformity in the process. The soil is conveyed by the rotary tillage knives and pick-up shovel via the scraper-type conveyor to the soil transport hood, which separates the soil into four paths. The soil on the sides of the membrane's edge is taken by the chute delivery, while the remaining soil slides down due to gravity and is discharged through the middle chute of the cross belt mulcher, completing the furrow's soil mulching process. In addition, as the combine moves forward with its operations, a portion of soil from the cross-waistband mulcher is discharged through the center-mounted chute. Residual soil keeps piling up, and the cross-belt mulcher and furrow are intermittently shaken through mutual contact of the cross-belt mulching opening mechanism and the ground wheel axle, facilitated by a belt-driven rotating eccentric wheel, to vibrate the soil buildup off the plate, allowing completion of transverse mulching at consistent spacing intervals in a full-cover dual-row mulched seed bed [18]. The main technical parameters of the initial mulching machine are shown in Table 1.

**Table 1.** Main technical parameters of the ridging and mulching machine.

| Parameters | Value/Mode |
|---|---|
| Overall size (L × W × H)/(m × m × m) | 1.78 × 1.50 × 1.05 |
| Matching power/kW | 18.4 |
| Hook-up method | Rear three-point suspension |
| Overall mass/kg | 166 |
| Operating speed/(km·h$^{-1}$) | 2.80~3.60 |
| Rotary plowing depth/mm | 100~150 |
| Membrane edge mulch width/mm | 90~110 |
| Ridge cover width/mm | 35~45 |
| Width of cross belt cover/mm | 90~110 |
| Distance between cross belt coverings/mm | 1500 |
| Mulch thickness/mm | 20~30 |

### 3. Wear Analysis Model

*Contact Model*

In this paper, EDEM was used to simulate the wear process of the cladding device. The Hertz–Mindin non-slip model was used to simulate interactions between sand and soil particles, as well as wear modeling of Archard wear between sand particles and the mulching unit. The model was used to assess the depth of wear of the cladding device, allowing effective prediction of the amount of material to be removed from the surface of the device [19]. Wear was determined using the relative wear model with accompanying data, for obtaining the normal and tangential cumulative contact energy of sand particles interacting with an overburden device and measuring the quantity of material removed at various points in the mulching apparatus.

Determining the wear constant using Archard wear theory can pose a significant challenge due to its complexity. Based on Archard wear theory, the wear volume of the workpiece surface $W_v$ can be quantified:

$$W_v = \partial_s \cdot F_n \cdot l, \tag{1}$$

$\partial_s$ is the wear constant, $l$ is the relative sliding distance of the particles in mm.

Combined with Figure 2, during the interaction between soil particles and the surface of the mulching device, define the wear volume on the surface of the cladding device as

$$W_v = \phi \cdot A_0 \cdot l \tag{2}$$

$\phi$ is the ratio of actual material removal to theoretical material removal, its size is 0.84.

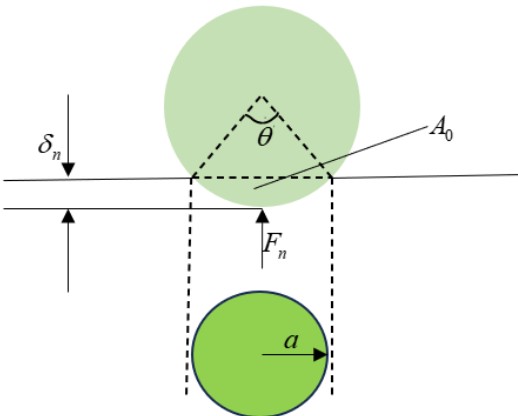

**Figure 2.** Schematic diagram of the action of soil particles on the surface removal of the mulching device. Notes: $a$ is the radius of the contact area, mm; $F_n$ is the reaction force on the particle, N; $\theta$ is the angle of the center of the corresponding arc, (°); $A_0$ is the cross-sectional area of the spherical indentation, mm$^2$; $\delta_n$ is the normal overlap, mm.

Combining Equations (1) and (2) results in

$$\partial_s = \frac{\phi \cdot A_0}{F_n} \tag{3}$$

As can be seen in Figure 2,

$$A_0 = \frac{\theta \cdot R_p^2}{2} - a(R_p - \delta_n) \tag{4}$$

$$a = \sqrt{R_p \delta_n} \tag{5}$$

$$\sin \frac{\theta}{2} = \frac{a}{R_p} \tag{6}$$

$R_p$ is particle radius, mm; $a$ is the radius of the contact area, mm.

From Equation (6),

$$\theta = 2\arcsin\left(\frac{\delta_n}{R_p}\right)^{\frac{1}{2}} \tag{7}$$

Combining (4)~(5) and Equation (7) gives

$$A_0 = \left[\arcsin\left(\frac{\delta_n}{R_p}\right)^{\frac{1}{2}} - \left(\frac{\delta_n}{R_p}\right)^{\frac{1}{2}} + \left(\frac{\delta_n}{R_p}\right)^{\frac{3}{2}}\right] R_p^2 \tag{8}$$

The contact force is calculated as

$$F_n = \frac{4}{3}E^* R_p^{\frac{1}{2}} \delta_n^{\frac{3}{2}} = \frac{4}{3}E^*\left(\frac{\delta_n}{R_p}\right)^{\frac{3}{2}} R_p^2 \tag{9}$$

$E^*$ is the equivalent modulus of elasticity.

The formulae acquired through the combination of Equations (3), (8) and (9)

$$\partial_s = \frac{3f\left[\arcsin\left(\frac{\delta_n}{R_p}\right)^{\frac{1}{2}} - \left(\frac{\delta_n}{R_p}\right)^{\frac{1}{2}} + \left(\frac{\delta_n}{R_p}\right)^{\frac{3}{2}}\right]}{4E^*\left(\frac{\delta_n}{R_p}\right)^{\frac{3}{2}}} \tag{10}$$

The relationship between particle hardness and yield stress can be expressed as

$$H_e = 3\sigma_c \tag{11}$$

$H_e$ is the hardness of the particles, Pa; $\sigma_c$ is the yield stress of the particle, Pa; the yield stress is approximately equal to the maximum compressive stress of the soil particles:

$$\sigma_c = P_m \tag{12}$$

$$P_m = \frac{2}{\pi}E^*\left(\frac{\delta_n}{R_n}\right)^{\frac{1}{2}} \tag{13}$$

can be obtained from Equations (11) and (13):

$$\frac{\delta_n}{R_p} = \left(\frac{\pi H_e}{6E^*}\right) \tag{14}$$

Substituting the sandy soil hardness ($4.2 \times 10^8$ Pa) into equation (14), the magnitude of the wear constant is obtained as $2.0 \times 10^{-12}$.

## 4. Discrete Elemental Modeling of Soil Wear for Mulching Devices-Seed Beds

### 4.1. Discrete Elemental Modeling of Cladding Devices

According to the full-film duopoly seed bed mulching requirements, the mulching device comprises two side-mounted chutes and soil transfer covers that are symmetrically linked using bolts on either side. The length of the side-mounted chute is 470 mm, the width 110 mm, a 100 mm depth should be measured at the root of the groove, the notch depth is 40 mm, and the thickness of the chute is 5 mm. The length of the soil transport housing is 450 mm, the width is 205 mm, the hood depth is 207 mm, and the shell thickness is 5 mm. Considering that an actual mulching device has more components and is symmetrical on both sides, a simplification of a mulching device was made in SolidWorks, simplifying the 3D model of the mulching device to a symmetrical half and eliminating parts that do not affect the simulation. The 3D model was saved in "x-t" format, due to the relatively coarse meshing in EDEM software, in order to better model the wear distribution on the surface of the cladding device, needed for grid refinement of mulching devices. The mesh module

in ANSYS Workbench 2020 R2 software was applied for triangular meshing of the model of the earth covering device, checking the quality of the grid after it had been divided, with a finalized grid size of 3 mm, where the soil transfer hood had 55,995 grids, while the chute had 17,304 grids. Figure 3 shows the meshing results for the mulching device.

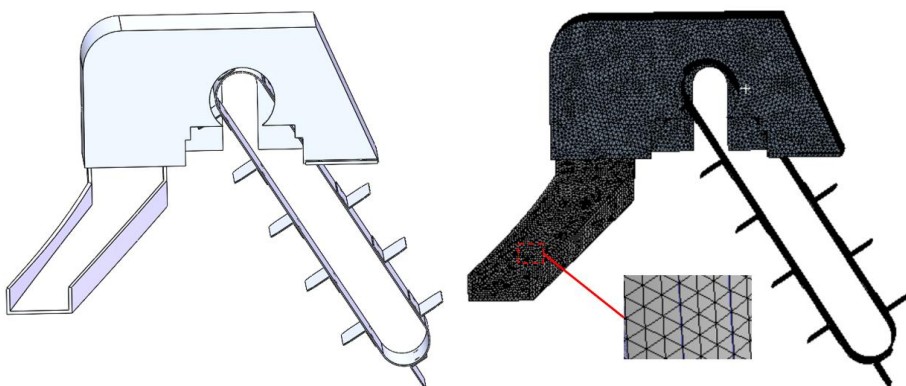

**Figure 3.** Mesh delineation results of the mulching device model.

### 4.2. Discrete Elemental Modeling of Sandy Soil Particles in Seed Beds

The establishment of a precise model for the sand particles in the seed bed was crucial for guaranteeing the accuracy of the simulation outcomes. Given that the particle size distribution is not uniform in actual sand particles, and to more closely resemble the actual sandy soil, non-standard spherical particles were used to simulate actual sandy soils. Existing studies have shown that the basic structure of sand particles includes three types: spherical, elongated, and prismatic, and the particle modeling was based on particles with a size of 1 mm. Their size was enlarged by a factor of 3, with spherical 1/2, elongated 1/3, prismatic 1/6, in EDEM. The basic parameters of each particle are shown in Table 2 [20].

**Table 2.** Basic parameters of the sand particle model.

| Typology | Spherical | Elongated | Prismatic |
|---|---|---|---|
| Single ball radius/mm | 1.25 | 0.9 | 0.9 |
| Mass/g | 75 | 50 | 25 |

In order to approximate the irregular shape of the particles, a standard spherical shape was chosen to fill the irregular shape [21]. For long stripes, they were filled with a linear array of standard balls; for the prismatic type, a standard spherical triangular array was selected to fill, based on the particle profile [22], as shown in Figure 4.

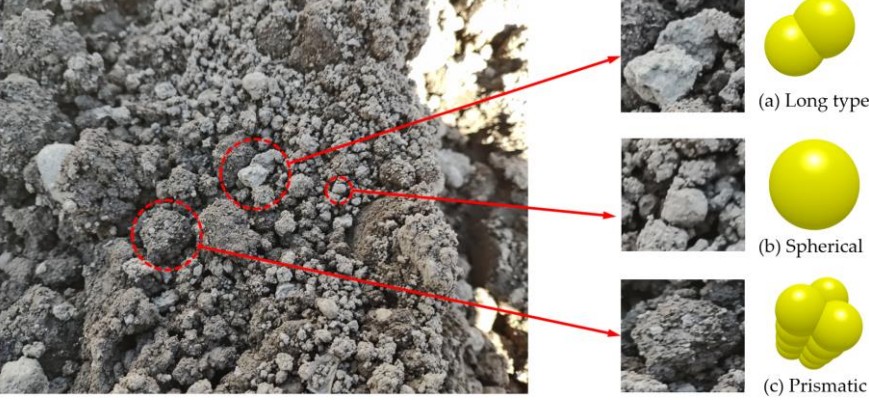

**Figure 4.** Discrete elemental model of three sandy soil particles.

According to the 'fall mulching' and 'top mulching' period, the soil water content is roughly between 10% and 20%, combining the characteristics of Huangmian sandy soil in arable land in the dry zone of Northwest China. Through experiments and a literature review, the sand and steel intrinsic parameters could be determined, as shown in Table 3 [20].

**Table 3.** Intrinsic parameters of particles and steel.

| Materials | Densities/(kg·m$^{-3}$) | Poisson's Ratio | Shear Modulus/Pa |
|---|---|---|---|
| Pellets | 1638 | 0.3 [23] | $1.15 \times 10^7$ |
| Steels | 7850 [24] | 0.25 [20] | $1.00 \times 10^{10}$ [20] |

*4.3. Discrete Element Model Contact Parameterization*

The seedbed soil consists of a large number of discrete sandy soil particles, and the mulching device and components cause severe wear and tear, due to contact with hard particles in the sand and soil particles during the mulching operation. In this paper, the soil covering device was used as a wear object, with simulation of the wear process of a mulching device using EDEM. As the water between sand particles and in the pore space forms a certain adhesive force due to surface tension [25], the Hertz–Mindlin with JKR contact model was used for the particle–particle contact model in EDEM [26]. This model was able to take into account the cohesion of the material on the basis of non-slip contact; meanwhile, the particle–geometry contact model in EDEM used the built-in relative wear model and Hertz–Mindlin with Archard wear model to simulate and analyze the wear of the operation process of the mulching device [27]. The relative wear model determines the impact wear and abrasive wear of sand particles on the cladding device, calculates the cumulative contact energy, and shows where the geometry is most susceptible to wear. The relative wear model measures four indicators of wear: tangential cumulative contact force, tangential cumulative contact energy, normal cumulative contact force, and normal cumulative contact energy. The Hertz–Mindlin with Archard ear model based on the J.F. Archard wear theory can give a specific wear value for the wear area of a mulching device [28]. The contact property parameters for the discrete element wear simulation of the cladding device are shown in Table 4 [20–29].

**Table 4.** Discrete element simulation parameters for wear process simulation.

| Parameters | Value |
|---|---|
| Coefficient of recovery for sand–sand collisions | 0.15 |
| Coefficient of recovery for collision between sand and an overburden device | 0.54 |
| Static friction factor of sand and sandy soil | 0.68 |
| Static friction factor between sand and an overburden device | 0.31 |
| Sand to sand rolling friction factor | 0.27 |
| Rolling friction factor between sand and mulch unit | 0.13 |

## 5. Simulated Wear Behavior Analysis of Cladding Devices

*5.1. Laws of Motion of Sand Particles in a Mulching Device*

In order to further study the abrasion mechanism of sand and soil particles on the mulching parts of the full-film double-row mulching device during the mulching operation process, numerical simulation was carried out of the device mulching operation process using the discrete unit method. For the sandy soil particles, in order to approximate the irregular shape of the particles, the standard sphere type was selected to model the irregular shape of the filled sphere particles, and the basic structure of the sand particles included three types: spherical, elongated, and prismatic. Particle modeling was based on particles with a size of 1 mm, scaled up by a factor of 3, 1/2 for spherical, 1/3 for elongated, and 1/6 for prismatic [20]. The simulation time step was 1. $405 \times 10^{-5}$ s, which is 40% of the Rayleigh time step, and the simulation was carried out for 2 s in total. For the scraper

lift belt lifter with 12 scrapers per side, the distance between the two scrapers was set to 100 mm, and the motion was controlled using dynamic coupling through the coupling server panel in EDEM. Based on the optimized values of the operating parameters of the mulching device, the forward speed was 0.7 m·s$^{-1}$, and the scraper lift belt lifter linear velocity was set at 0. 67 m·s$^{-1}$ [18]. The particle plant was a 135 mm × 100 mm rectangular plane, and the particle plant generated sandy soil particles at a rate of 3.5 kg·s$^{-1}$.

The velocity change process of sand particles during mulching with the mulching device could be obtained with the EDEM post-processing module. After *t* = 1.84 s, the mulching device tended to stabilize the mulch delivery state, and the thickness of the resulting sand particle stream was relatively uniform and consistent. Therefore, Figure 5a–f show the movement processes and patterns of sand and soil particles within the cladding device during the time period *t* = 0–1.84 s.

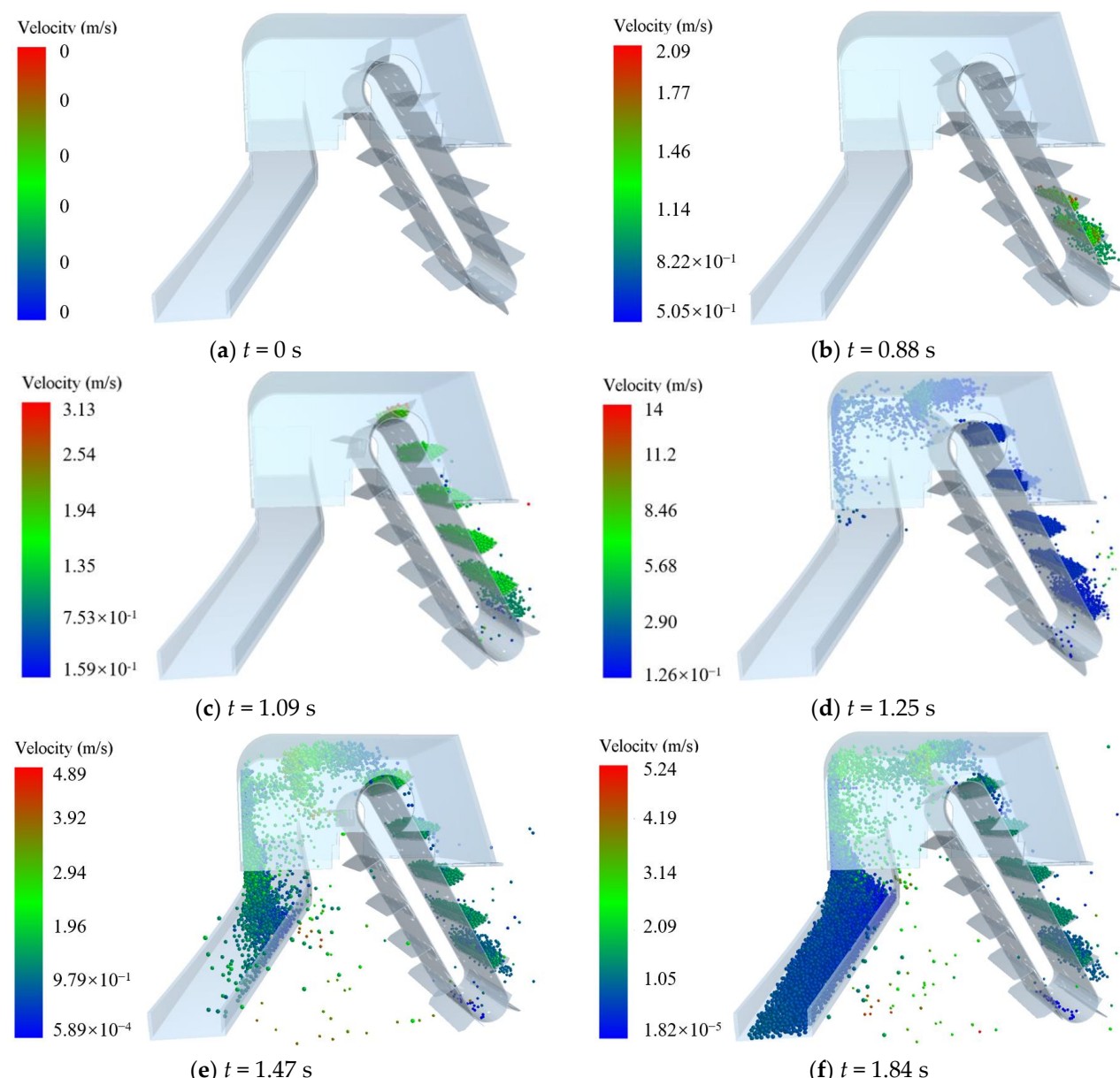

**Figure 5.** The movement process and law of sand particles in the mulching device.

When *t* = 0 s, the lift belt lifter scraper started moving at a 0.67 m·s$^{-1}$ [18] linear velocity, and a steady inclined lift line speed was reached within (0~0.8) s (Figure 5a); when *t* = 0.8 s, the particle plant started to generate spherical, elongated, and prismatic sand

particles with an initial velocity of $-1$ m·s$^{-1}$ in the $y$-axis direction at a generation rate of 3 kg·s$^{-1}$, where the ratio of particles was 1/2 for spherical, 1/3 for elongated, and 1/6 for prismatic [20]. The sand and soil particles reached a velocity of 1.14 m·s$^{-1}$ under their own gravity and the action of the lifting shovel, to enter the lift belt lifter and then contact the scraper and fill (Figure 5b). When simulating a time range of 1.09–1.25 s, the power shaft rotation of the lifting belt lifter drove the scraper conveyor. This process allowed for the inclined lifting and conveying of the sand and soil particles, which can be seen in Figure 5c. The sand and soil particles were lifted and conveyed at an inclined angle for further processing. The sand particles filled into the scraper conveyor and stable inclined lift between the two scrapers and the filled sand particles, due to the scraper extrusion and friction under the action of the formation of a closed right-angle triangle, and this meant that the sand and soil particles above the scraper conveyor had filled to a sufficient quantity and begun to gradually enter the soil conveying shell at a speed of 1.94 m·s$^{-1}$. The sand and soil particles were impacted at the bottom surface of the shell of the soil transport housing by the lifting action of the scraper at a maximum speed of 5.68 m·s$^{-1}$. With impact, friction, crushing, and scraping of sand particles on the bottom and left side surfaces of the soil transport casing due to their own inertia and surface irregularities, with the maximum abrasion damage of particles on the soil transport casing. After the sand particles had passed through the soil transport casing, the velocity of the particles decreased to 2.9 m·s$^{-1}$ under the reverse interception force of the casing and its own gravity. At this point, a small, discontinuous stream of sand particles occurred at the skid chute of the overburden device (Figure 5d). When the simulation time reached 1.47~1.84 s, the uplifted sand particles passed through the soil cover of the mulching device, the velocity of the sand particles increased to 4.89 m·s$^{-1}$ under the interaction of the sand particles with the soil transport hood and the gravity of the sand particles themselves, and the movement of sand and soil particles into the chute drop zone had a greater impact on this area. At this point, the normal contact accumulation energy reached its peak, and the wear damage to the skidding trough was the most severe (Figure 5e). After passing through the drop zone of the chute, the sand particles were intercepted by the 75° overburden side channel angle side plate, and then the particle velocity decreased to 1.05 m·s$^{-1}$. At this stage, the sand particles were scratched and worn by their own gravity and the sharp corners and edges of the sand particles on the chute, and the sand particles in the chute gradually formed a coherent flow of sand particles. The flow of sand particles gradually increased and tended to stabilize the overburden transport state (Figure 5f). The amount of mulch on mulched seed beds is a key factor influencing the functional stability of a full-film duopoly production system; too much or too little mulch can have an effect on the effectiveness of the seed bed construction. When the amount of mulch is too large, the effective light area of the mulched seed bed is reduced, with serious constraints on the production function of "ground temperature enhancement and rainwater harvesting on the membrane surface". When the amount of mulch is too small, the mulched seed beds are less likely to settle close to the surface, making it difficult to withstand the natural winds of the outside world and removing the film, which results in the failure of the "cover and vapor suppression" function of the seed bed [30]. Therefore, in order to ensure the consistency and stability of the mulching of mulched seedbeds during the operation of the soil transfer-bed mulching device of the operating machine, calculations need to be made of the amount of overburden. Combining the construction characteristics of full-membrane double-row ridged seed beds in Gansu Province, the horizontal belt mulching type full-film double-monopoly furrow mulching machine main technical parameters could be determined, as shown in Table 1: laminator operating speed 2.80–3.60 km·h$^{-1}$, membrane edge mulch width 90~110 mm, and mulch thickness of 20~30 mm, and the theoretical soil cover at the membrane side of the chute was calculated to be 3~4 kg·s$^{-1}$. When the simulation time reached 1.84 s, the mulching device tended to stabilize the mulch delivery state. To clarify the operational performance of the mulching device, a sensor was installed at the outlet of the mulching device chute, for detection of the amount of mulch after the action of the mulching device and calculation

of mulch results from sensor data. The soil cover at the outlet of the skidding chute was 3.46 kg·s$^{-1}$. Therefore, from the comparison of the theoretical mulching amount and the mulching amount obtained from the sensor detection, it can be seen that the mulching device mulched the soil better.

### 5.2. Trajectory of Sand Particles in the Mulching Device

In the discrete element mulching simulation, the lifting line speed of the scraper conveyor was adjusted to ensure that the sand particles were discharged from the bottom outlet of the chute after mulching. In order to further improve the mulching performance of the mulching device, and to investigate the movement law of sand and soil particles during mulching operation, the trajectories of three different types of sand particles during the overburden process were labeled in the EDEM post-processing module, as shown in Figure 6a, where the blue trajectory line represents the movement of spherical particles, the green trajectory line represents bar particles, and the red trajectory line represents the movement of prismatic particles. The sand and soil particles entered the lift belt lifter under their own gravity and the action of the lifting shovel and then came into contact with the scraper and were filled. As the power shaft of the lifting belt-type soil lifter rotated to drive the scraper-type conveyor, it realized the inclined lifting and conveying of sand and soil particles into the soil conveying shell. The sand and soil particles under the lifting action of the scraper impacted at high speed on the soil transporting cover, which produced impact wear areas on the soil transporting cover, as shown in Figure 6b. After the sand particles passed through the chute drop zone and were intercepted by the 75° soil-covered side-flow chute angle side plate, the sand particles in the chute gradually formed a coherent flow of sand particles, and the gravity of the sand particles and the sharp corners and edges of the sand particles scratched the chute to produce a scratched wear area, as shown in Figure 6b.

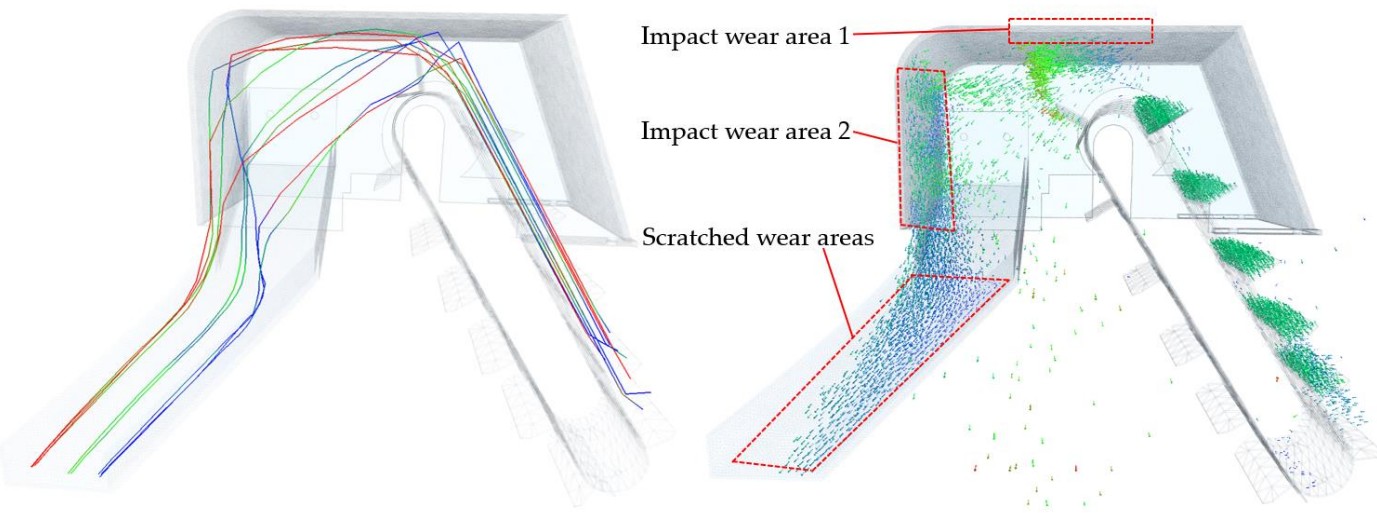

(**a**) Trajectory of sandy soil particles          (**b**) Areas of wear and tear on mulching devices

**Figure 6.** Trajectory of sand particles in the mulching device and wear areas.

### 5.3. Analysis of Cladding Device Wear during the Cladding Process

After the mulching device had completed the discrete element mulch wear simulation operation, EDEM's built-in relative wear model was utilized to identify the wear location of the overburden device and to determine the distribution of the cumulative contact energy of the wear. EDEM's built-in Hertz–Mindlin with Archard wear model was used to determine the specific actual wear depth of the overburden device, as shown in Figure 7.

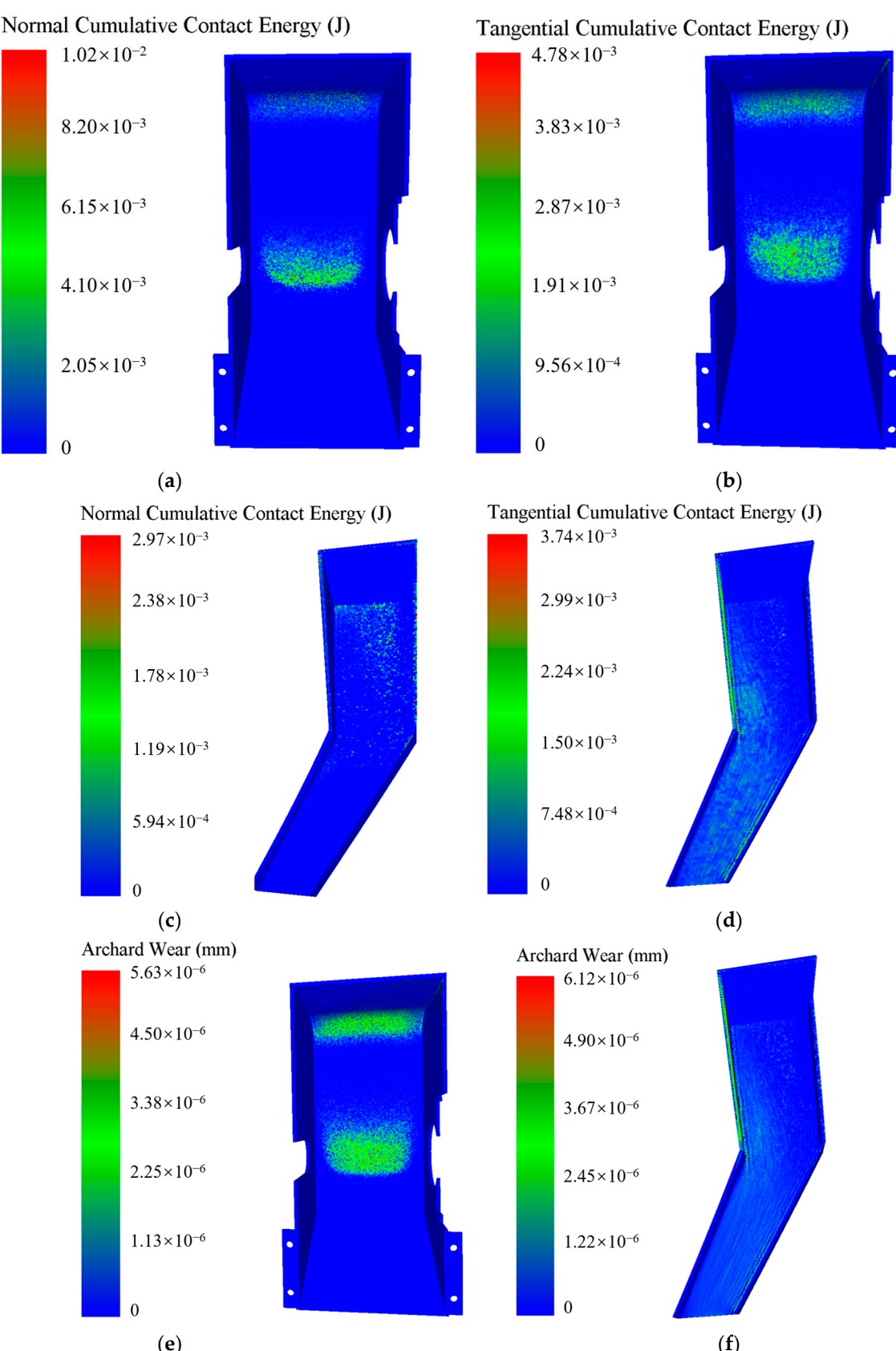

**Figure 7.** Distribution of wear areas in the mulch unit. (**a**) Cumulative contact energy normal to the soil transport casing. (**b**) Tangential cumulative contact energy of the soil transport casing. (**c**) Cumulative contact energy normal to the chute. (**d**) Tangential cumulative contact energy of the chute. (**e**) Depth of wear and tear of the soil transport casing. (**f**) Depth of wear of chute.

Figure 7a,b show the cumulative contact energy of wear in the normal and tangential directions, respectively, of the soil transport casing, which can reflect the wear locations of the soil transport casing. As the sand and soil particles impacted on the bottom surface of the shell of the soil transfer casing at a speed of 5.68 m·s$^{-1}$ under the lifting action of the scraper, normal impact, friction, extrusion, and tangential scraping and scratching of the bottom and left side surfaces of the earth transfer casing by sand particles occurred, due to their own inertia and surface irregularities. Therefore, the cumulative contact energy in the normal direction was greater than the cumulative contact energy in the tangential direction, the interaction between the surface of the soil transport casing and the sand particles was intense, and the cumulative contact energy was more concentrated. The wear contact energy was concentrated on the bottom surface of the cowling and the left side shell surface, and the maximum value of the normal cumulative contact energy was $1.02 \times 10^{-2}$ J, while the maximum value of tangential cumulative contact energy was $4.78 \times 10^{-3}$ J. There was more severe wear on the soil transport housing, with a maximum wear depth of $5.63 \times 10^{-6}$ mm, and the depth of wear on the bottom surface of the shell of the soil transport housing was greater than the depth of wear on the left side of the surface. The bottom surface of the shell of the soil transfer casing was the main working part taking the impact of the sand and soil particles, and it had the most intense interaction with the sand and soil particles. The velocity of the sand particles reduced to 2.9 m·s$^{-1}$ due to the collision with the bottom surface of the shell of the soil transport casing, and this resulted in the left surface of the soil transport casing being subjected to less wear than the bottom surface of the soil transport casing, as shown in Figure 7e.

Figure 7c,d show the cumulative contact energy of wear in the normal and tangential directions of the chute, respectively, which reflects the distribution of the wear regions of the chute, where the tangential wear cumulative contact energy of the chute was greater than the normal wear cumulative contact energy. As the uplifted sand particles passed through the soil transport housing of the mulching device, the sand particles were transported to the soil transport housing. Under the interaction of the sand particles with the soil transport hood and the gravity of the sand particles themselves, the velocity of the sand particles decreased to 4.89 m·s$^{-1}$. The movement of sand and soil particles into the chute drop zone had a greater impact on this area. At this point, the normal wear contact accumulation energy reached its peak value, and the wear areas were mainly concentrated in the tributary area at the bottom of the chute. The maximum value of cumulative contact energy for normal wear was $2.97 \times 10^{-3}$ J, and the wear and tear damage to the chute was most severe. After passing through the drop zone of the chute, the sand particles were intercepted by the 75° overburden side-flow chute angle side plates, and then the particle velocity continued to decrease to 1.05 m·s$^{-1}$, which was the lowest velocity of the chute. At this stage, the sand particles were scratched and worn by their own gravity from the sharp corners and edges of the sand particles on the chute, and the sand and soil particles in the chute gradually formed a coherent flow, and the flow of sand and soil particles gradually increased to a stable state of mulch transportation. At this point, the cumulative contact energy of tangential wear reached its peak value, and wear areas were located in the chute exit area, the shape of the wear marks on the surface of the chute was long, and the different wear marks were distributed in parallel. The maximum value of cumulative contact energy for tangential wear was $3.74 \times 10^{-3}$ J. That the wear of the chute was less than the wear of the conveyor housing, and that the depth of wear in the upper drop zone of the chute was greater than the depth of wear on the surface of the lower deflector zone of the chute, was mainly due to the fact that the sand particles were lifted from the soil lifter into the soil transport casing, the velocity of sand and soil particles decreased after the two contacted each other, and entered the chute to form a stable particle flow. The scratch wear of the chute surface by the surface corners of the sand particles had a maximum wear value of $3.67 \times 10^{-6}$ mm, as shown in Figure 7f.

*5.4. One-Factor Simulation Test*

The main function of a scraper conveyor is to lift and transport the sand particles dug up by the earth-moving shovel through the scraper lifting belt to the conveying hood, and move the sand particles through the soil transport cover shell with contact interaction into the chute, to ensure that the chute has sufficient mulching sand particles on the seed bed film edge and furrow uniformity, for continuous mulching operations. Therefore, a reasonably selected range of scraper conveyor lifting line speeds is critical to the effectiveness of mulching. Based on the results of the previous design tests, the inclination angle of the scraper conveyor was 45° [31], the scraper conveyor filling factor was 0.80 for the inclined lift [32], the angle of internal friction of sandy soil was 28° [33], the laminator operating speed was 2.80–3.60 km·h$^{-1}$, the membrane edge mulch width was 90~110 mm, the mulch thickness was 20~30 mm, the calculated theoretical soil cover at the membrane side of chute was controlled at 2.5~4.5 kg·s$^{-1}$, and the calculated scraper lift belt lifter scraper spacing was control within 80~120 mm. Too slow a speed affects the amount of mulch, and too fast a speed creates too much dust and too much pollution. Comprehensive consideration determined that the scraper conveyor lifting line speed needs to be controlled at 0.7~1.5 m·s$^{-1}$. EDEM 2022.2 software was used to simulate the wear and tear of the mulching device under different test factors, and the corresponding test data were obtained, in order to select a superior level of factors. The amount of wear and tear was used as the test indicator, and a one-factor simulation test was conducted using the conveyor lift line speed of the mulcher, the amount of mulch in the seedbed, and the spacing of the scrapers as the factors. The results are shown in Figure 8.

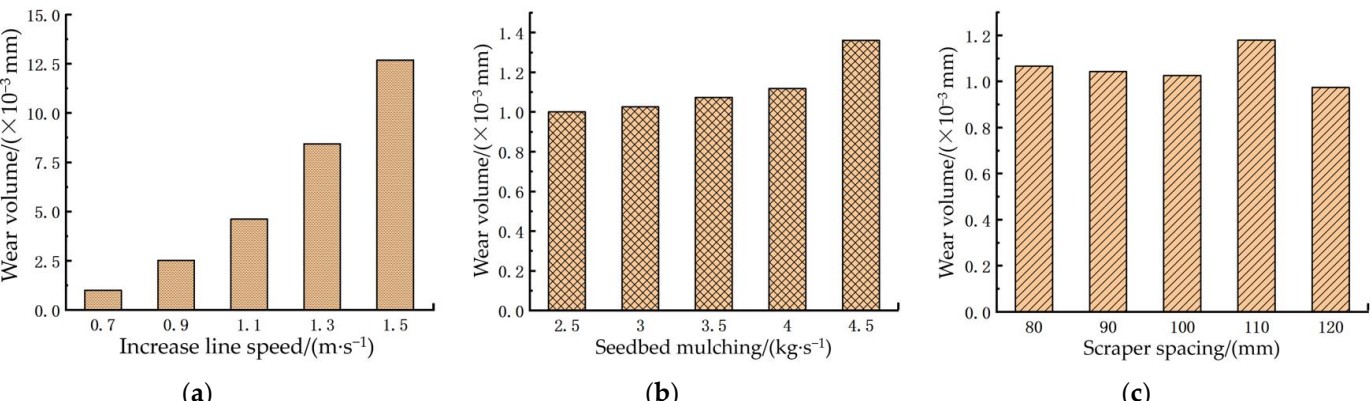

**Figure 8.** Results of the one-factor test. (**a**) Impact of increased line speed on mulching device wear. (**b**) Effect of seed bed mulch on mulching device wear. (**c**) Influence of scraper spacing on the amount of wear on the mulching device.

As can be seen from Figure 8a, when the spacing of the scrapers was 100 mm and the amount of soil covering on the seed bed was 3 kg·s$^{-1}$, as the lifting line speed increased, the scraper conveyor lifted the overburden sand particles thrown by the soil lifting shovel at higher line speeds into the soil transport housing, due to the gradual increase in the scraper conveyor line speed from 0.7 m·s$^{-1}$ to 1.5 m·s$^{-1}$. The impact of the sand and soil particles on the soil transport casing due to their own inertia gradually intensified, leading to increased impact wear on the bottom surface of the soil transport casing, where the impact of the sand and soil particles on the chute was aggravated by the force of gravity when they fell into the chute drop zone after the contact operation of the soil conveyor shell. The irregular edges of the surface of the sand particles increased the scratching and wear on the chute guide area, so there was a linear increase in the amount of wear on the mulching device, with a minimum wear of $1.001 \times 10^{-3}$ mm at a lifting line speed of 0.7 m·s$^{-1}$, while the maximum amount of wear of the cladding device was $12.51 \times 10^{-3}$ mm when the lifting linear velocity was 1.5 m·s$^{-1}$. As can be seen from Figure 8b, when the scraper spacing was 100 mm and the lifting line speed was 0.7 m·s$^{-1}$, as the amount of mulch in

the seed bed increased, the amount of sand particles lifted up by the scraper conveyor gradually increased, with a linear increase in cumulative contact energy of sand and soil particles normal to the transport casing wear. The cumulative contact energy of tangential abrasion on the soil transport casing due to the gravity of the sand particles passing over the left surface of the casing increased linearly with the amount of overburden. When the sand particles flowed through the backflow area of the chute, the chute wear tended to increase gradually, and a minimum abrasion of $1.026 \times 10^{-3}$ mm was observed at a seed bed mulch rate of 2.5 kg·s$^{-1}$, while the maximum wear of the mulching device was $1.396 \times 10^{-3}$ mm when the seed bed was mulched at 4.5 kg·s$^{-1}$. From Figure 8c, when the lifting linear velocity was 0.7 m·s$^{-1}$ and the seed bed mulching volume was 3 kg·s$^{-1}$, and when the scraper spacing was gradually increased from 80 mm to 120 mm, there was a reduction in the number of sand particles lifted per unit of time by the scraper type soil lifter. The accumulated energy of normal impact wear on the soil transport casing gradually reduced, decreasing in the total number of sand and soil particles passing through the soil transport casing. As a result, the cumulative contact energy of tangential abrasion on the chute from sand particles flowing through the chute was reduced. Therefore, the wear of the mulching device tended to decrease gradually, with the minimum wear at a 120 mm scraper spacing of $0.974 \times 10^{-3}$ mm, while the maximum wear of the mulching device was $1.216 \times 10^{-3}$ mm when the scraper spacing was 110 mm.

## 6. Numerical Simulation Optimization Test

### 6.1. Box–Behnken Experimental Design

Mulching unit group wear was evaluated using EDEM software for various test factors (scraper lift line speed, seedbed mulch volume, and scraper spacing), to obtain the corresponding test data, applying Design-Expert 8.060 software to analyze the influence pattern of the above factors on the cladding device. To further analyze the effects of the scraper conveyor lifting line speed, seed bed cover, scraper spacing, and their interactions on performance, based on the results of the one-way test, a Box–Behnken central combination test was used to explore the influence of each parameter on the wear performance of the mulching device and to find the optimal parameters for the mulching device. With development of a three-factor, three-level centralized test, the factor levels were coded as shown in Table 5, and a total of 17 sets of response surface analysis tests were performed.

**Table 5.** Experimental factors and coding.

| Experimental Factors | Coding Level | | |
|:---:|:---:|:---:|:---:|
| | **−1** | **0** | **1** |
| Lifting line speed $X_1$/(m·s$^{-1}$) | 0.7 | 0.9 | 1.1 |
| Amount of soil cover $X_2$/(kg·s$^{-1}$) | 2.5 | 3 | 3.5 |
| Blade spacing $X_3$/mm | 90 | 100 | 110 |

Notes: $X_1$, $X_2$, and $X_3$ are the horizontal values for lift line speed, seedbed cover, and scraper spacing, respectively.

### 6.2. Test Results and Analysis

#### 6.2.1. Regression Modeling and Testing

The numerical simulation results are shown in Table 6. The test results were analyzed using Design-Expert 8.060 software, and a quadratic polynomial regression was applied to the experimental data in order to remove insignificant factors. A regression equation was established between the lifting line speed $x_1$, seed bed mulch $x_2$, scraper spacing $x_3$, and mulching device wear $Y$, as shown in Equation (15). A quadratic regression model was obtained for the amount of wear $Y$ of the mulching device:

$$Y = 2.53 + 1.72x_1 + 0.2575x_2 + 0.1925x_1x_2 - 0.1478x_1x_3 + 0.1475x_2x_3 + 0.2489x_1^2 \quad (15)$$

**Table 6.** Experimental design scheme and results.

| Serial No. | $x_1$ | $x_2$ | $x_3$ | Mulching Unit Wear/($\times 10^{-3}$ mm) |
|---|---|---|---|---|
| 1 | −1 | 0 | 1 | 1.18 |
| 2 | 0 | 0 | 0 | 2.64 |
| 3 | 0 | 1 | −1 | 2.62 |
| 4 | 1 | 0 | 1 | 4.38 |
| 5 | 0 | −1 | 1 | 2.22 |
| 6 | 0 | 0 | 0 | 2.51 |
| 7 | −1 | 0 | −1 | 1.07 |
| 8 | 0 | 0 | 0 | 2.48 |
| 9 | 0 | −1 | −1 | 2.34 |
| 10 | 0 | 0 | 0 | 2.53 |
| 11 | 1 | 0 | −1 | 4.86 |
| 12 | −1 | 1 | 0 | 1.06 |
| 13 | 1 | 1 | 0 | 4.84 |
| 14 | 1 | −1 | 0 | 4.00 |
| 15 | 0 | 0 | 0 | 2.49 |
| 16 | 0 | 1 | 1 | 3.09 |
| 17 | −1 | −1 | 0 | 0.99 |

6.2.2. Analysis of Variance of Regression Equations

Analysis of variance (ANOVA) was conducted on the experimental data using Design Expert software, and the results of the significance test of the model and regression coefficients are shown in Table 7.

**Table 7.** ANOVA of regression equations.

| Variation Source | $S_s$ | $d_f$ | $M_s$ | $F$ | $p$ |
|---|---|---|---|---|---|
| $x_1$ | 23.7300 | 1 | 23.7300 | 1775.4200 | <0.0001 ** |
| $x_2$ | 0.5304 | 1 | 0.5304 | 39.6800 | 0.0004 ** |
| $x_3$ | 0.0000 | 1 | 0.0000 | 0.0034 | 0.9553 |
| $x_{1\times2}$ | 0.1482 | 1 | 0.1482 | 11.0900 | 0.0126 * |
| $x_{1\times3}$ | 0.0873 | 1 | 0.0873 | 6.5300 | 0.0378 * |
| $x_{2\times3}$ | 0.0870 | 1 | 0.0870 | 6.5100 | 0.0380 * |
| $x_1{}^2$ | 0.2608 | 1 | 0.2608 | 19.5100 | 0.0031 ** |
| $x_2{}^2$ | 0.0131 | 1 | 0.0131 | 0.9834 | 0.3544 |
| $x_3{}^2$ | 0.0367 | 1 | 0.0367 | 2.7500 | 0.1414 |
| Model | 24.9000 | 9 | 2.7700 | 206.9700 | <0.0001 ** |
| Residual | 0.0936 | 7 | 0.0134 | | |
| Incoherent | 0.0770 | 3 | 0.0257 | 6.1800 | 0.0554 |
| Inaccuracies | 0.0166 | 4 | 0.0042 | | |
| Sum | 24.9900 | 16 | | | |
| Coefficient of determination (CoD) $R^2$ | 0.9963 | | | | |
| Research $R^2$ | 0.9914 | | | | |

Notes: $S_s$ is sum of squares; $d_f$ is degree of freedom; $M_s$ is mean squares; * means significant ($p < 0.05$); ** means extremely significant ($p < 0.01$).

In Table 7, $p < 0.0001$ for the regression equation, it is shown that the obtained quadratic regression model of mulching device wear was extremely significant, with a coefficient of determination $R^2 = 0.9963$ and good fit of the model. With $R^2 = 0.9914$ showing a high correlation between the predicted and tested values, the model could be used to analyze and predict the amount of wear on the mulching device. The misfit term $p = 0.0554 > 0.05$, and the misfit was not significant, indicating that the quadratic regression equation fitted by the model was consistent with the numerical simulation test results, where unknown factors had a small effect on the results of the test. Correctly reflecting the relationship between the amount of wear on the mulching device, $Y$, and $x_1$, $x_2$, and $x_3$, the regression model could best predict the various test results in the optimization test. Where the $x_1$, $x_2$, and $x_1{}^2$ of the model had highly significant effects on the response values, with $p$ values

were less than 0.01; $x_{1\times2}$, $x_{1\times3}$, and $x_{2\times3}$ had a significant effect on response values, with *p* values less than 0.05; while $x_3$, $x_2{}^2$, and $x_3{}^2$ had no significant effect on the response values. This shows that the effect of the test factors on the amount of wear of the mulching device was not a simple linear relationship and had an interaction effect. From the magnitude of the regression coefficients of the factors of the model, the order of predominance of the effects of the factors was $x_1$, $x_2$, $x_3$; that is, lifting line speed $x_1$, seed bed cover $x_2$, and scraper spacing $x_3$.

6.2.3. Parsing of Model Interaction Terms

In order to analyze the degree of the relative effect of the different interaction levels of the test factors on each indicator, based on the obtained quadratic regression model, response surfaces were plotted for the different interaction levels of lifting line speed, seed bed cover, and scraper spacing on the amount of wear of the cover device (Figure 9).

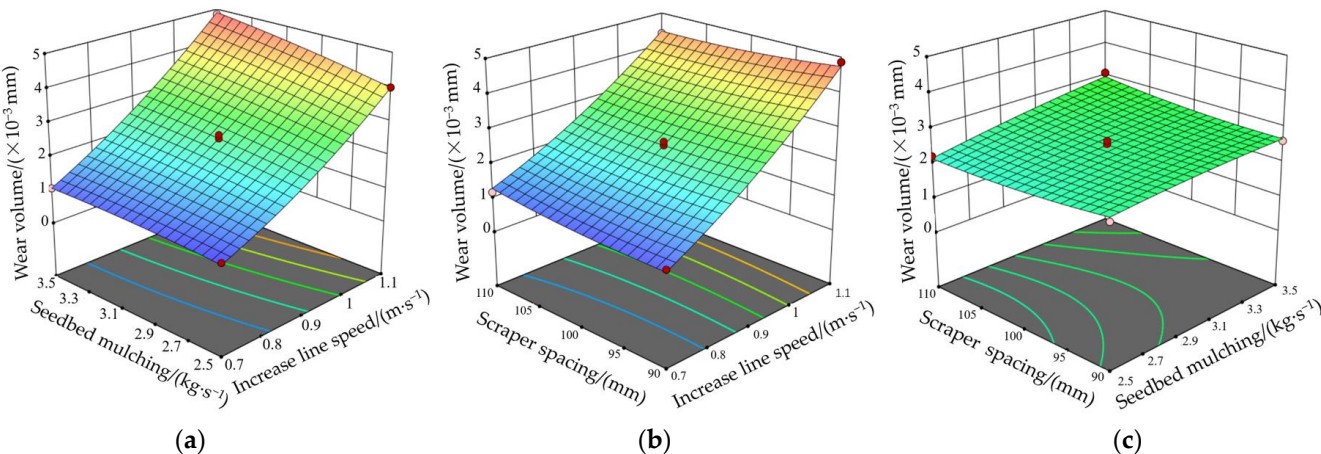

**Figure 9.** Response surface of wear volume of mulching device. (**a**) Influence of lifting line speed and seed bed cover on abrasion amount. (**b**) Effect of lifting line speed and scraper spacing on wear. (**c**) Influence of seed bed cover–scraper spacing on the amount of abrasion.

By exploring the effect of the interaction between these three factors on the amount of wear on the mulching device, the degree of wear on the mulching device by sand particles during mulching operations could be predicted. As can be seen from Figure 9a, when $x_3$ was at a zero level, the contours between the factors $x_1$ and $x_2$ were tight, indicating a significant interaction. The cladding device wear increased significantly with $x_1$ and then decreased with $x_2$. When the scraper spacing was certain and the lifting line speed of the cladding device was increased from 0.7 m·s$^{-1}$ to 1.1 m·s$^{-1}$, the wear of the cladding device increased linearly. At the same time, when the mulching volume of the seed bed of the mulching device was increased from 2.5 kg·s$^{-1}$ to 3.5 kg·s$^{-1}$, the wear of the mulching device increased slowly to a maximum value and then decreased slowly. It is known that lifting linear velocity has a highly significant effect on the amount of wear, and the amount of seed bed cover has a significant effect on the amount of wear. From Figure 9b, when $x_2$ was at zero level, the contours between the factors $x_1$ and $x_3$ were tight, indicating a significant interaction. The cladding device wear increased significantly with $x_1$, and the cladding device wear decreased and then increased with $x_3$. When the amount of seed bed mulch was certain, the amount of mulch device wear increased significantly as the lifting line speed increased. When the lifting line speed was stabilized at a certain value, as the scraper spacing was increased from 90 mm to 110 mm, there was no significant change in the amount of wear of the covering device, indicating that the influence of scraper spacing on the amount of wear of the covering device was not significant. From Figure 9c, when $x_1$ was at a zero level, the contours between the factors $x_2$ and $x_3$ were tight, indicating a significant interaction. When the lifting linear speed was certain, the scraper spacing had no significant effect on the amount of wear of the mulching device, while the amount of

wear of the mulching device gradually increased with the increasing amount of mulch in the seed bed. A simulated mulch device wear test targeting minimal wear was conducted. Optimizing the results with Design Expert optimization, the optimal parameters for the mulching device were a lifting line speed of 0.7 m·s$^{-1}$, seed bed mulching volume of 2.55 kg·s$^{-1}$, and scraper spacing of 98 mm. Under these conditions, the minimum wear of the mulching device was $0.958 \times 10^{-3}$ mm.

Comparison was made of the simulation results of the mulching device and the wear of sand particles under normal and optimal operating parameters, using trials 11 and 16 of the factorial test as examples, the resulting wear cloud is shown in Figure 10.

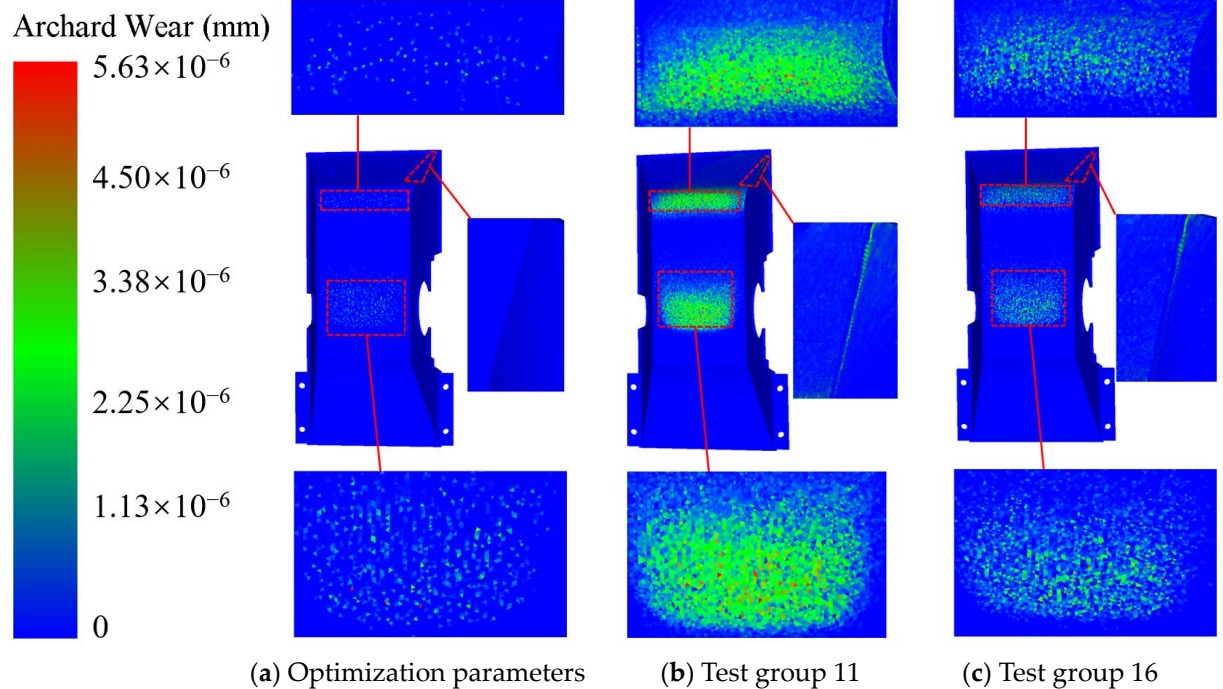

(**a**) Optimization parameters      (**b**) Test group 11      (**c**) Test group 16

**Figure 10.** Cloud comparison of wear effect of the mulching device.

Under the optimal operating parameters, the wear of the soil covering device and sand particles was mainly concentrated on the bottom impact surface of the soil transport hood shell and less on the left side surface. Under the general operating parameters for the impact wear between the soil covering device and sand particles, in addition to the concentrated wear on the impact surface of the bottom of the soil transfer casing, the wear area was distributed over the entire surface of the soil transfer casing, and there was a certain degree of damage to the shape and structure of the soil transfer casing. Thus, it can be concluded that the wear of the mulching device caused by mulching of the mulching combine under general operating parameters was greater than the wear under the optimal operating parameters, and there was also some damage to the profile structure of the mulching device. It can be judged that when the mulcher mulched the soil using the optimal working parameters, the abrasion and damage effect of sand and soil particles on the mulcher was smaller than that of the general working parameters, which can guarantee the mulching quality of the mulcher and prolong the working life of the mulching combination to a certain extent.

## 7. Field Validation Tests

To further verify the abrasion effect of sand and soil particles on the mulching device of a full-film dual-row ridging and mulching combine under optimal structural parameters, in June 2023, a field trial was carried out at the test field of the Taohe Tractor Manufacturing Co. The water content of the sandy soil in the test site was 16.86 percent, the sandy soil

capacity was 1300 kg·m$^{-3}$, the firmness was <0.20 MPa, with wider and looser fields with few previous crops. The field tests revealed that the combined machine operating in soil with high solidity, sand particles, and stones, as well as on uneven surfaces, was prone to damage of the cover shell of the mulching device and surface wear and tear of the skidding chute, leading to poor mulching outcomes. The study mainly focused on the prediction of wear loss sites through discrete elemental simulation of wear cladding operation processes using cladding devices, as well as reduction in the wear of the mulching device through optimizing the structural parameters of the device. The test wear effects are shown in Figure 11.

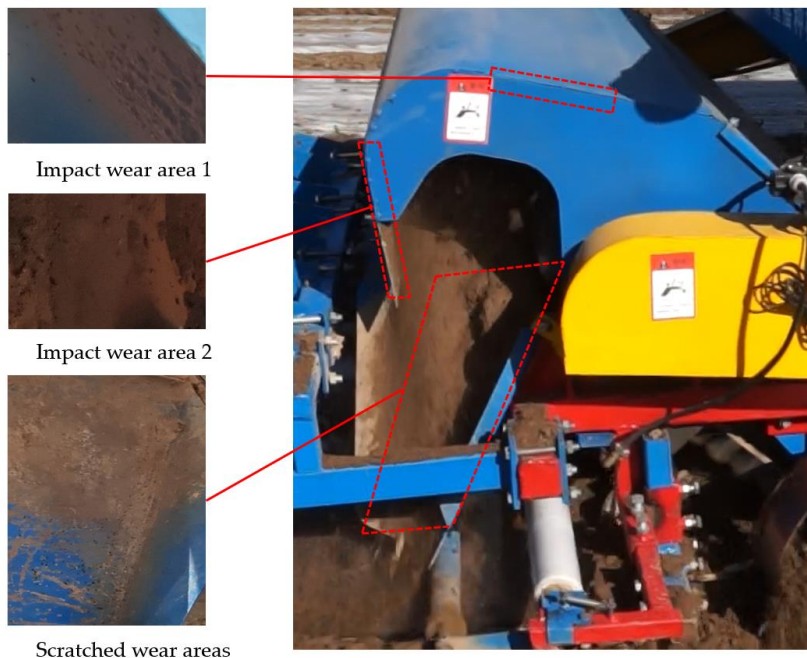

**Figure 11.** Field test of the wear of the mulching device.

A comparison of the simulation of the wear parts and the prototype of the actual mulching operation found that the two worn parts were the same, indicating that the establishment of the mulching device discrete element wear simulation model and structural parameters were reasonable, with a feasible abrasion mechanism process of sand particles on soil-covering devices. The simulation of mulch wear in a full-film two-monopoly furrow aligned with the results of the field mulch wear verification test, demonstrating the reliability and effectiveness of the model.

## 8. Conclusions

(1) For the operational characteristics of the mulching device of a dryland full-film dual-monopoly furrow ridging and mulching combine, a mulching wear model of the mulching device and mulching sand particles was established. The main forms of wear in the operation of the mulching device were analyzed to be impact wear and scratch wear. Combined with the agronomic requirements of mulching in full-film double-row seed beds, EDEM discrete element software was used to establish a three-dimensional model of the mulching device and the sandy soil particles of the double-row seed beds, to realize the mulching process. The cladding wear process between the cladding device and the cladding sand particles was analyzed, and two areas of impact wear on the overburden conveyor housing and areas of wear on the chute deflector scratches were detected.

(2) Combined with a Box–Behnken experimental design principle, a three-factor and three-level response surface analysis method was adopted to carry out a simulation test of mulching device mulch wear under different working parameters, where quadratic regression modeling of the soil transport casing and chute of the overburden device was performed with the help of Design-Expert 8.060 software. The effects of the scraper lifter

lifting line speed, seed bed cover, and scraper spacing on the amount of wear on the mulching device were analyzed, obtaining the optimal combination of operating parameters to minimize the wear of the mulching device. They were a lifting line speed of $0.7 \text{ m·s}^{-1}$, seed bed cover of $2.55 \text{ kg·s}^{-1}$, and scraper spacing of 98 mm. Under these conditions, the minimum wear of the mulching device was $0.958 \times 10^{-3}$ mm.

(3) Verification of the optimal working parameters of the resulting mulching machine was carried out through practical field tests, with a sandy soil at 16.6% water content. After optimization of the operating parameters of the mulching device, the results of the field trial validation, using a comparison of the simulation of the wear parts and the prototype for the actual mulching operation of the same two wear parts, showed that the discrete element wear simulation model and structural parameters of the established mulching device were reasonable, with a feasible abrasion mechanism process of sand particles on soil-covering devices. The simulation of the full-film dual-monopoly furrow mulch wear process aligned with the field mulch wear verification test, suggesting the model's dependability and efficiency. It can provide a reference for the mechanized full-film double-monopoly furrow seed bed mulching operation mode and for the optimization and loss reduction of the mulching device of mulching machines.

**Author Contributions:** Conceptualization, Q.Z.; methodology, Q.Z.; software, Q.Z.; validation, Q.Z., F.D. and P.X.; formal analysis, Q.Z.; investigation, R.S. and H.P.; resources, F.D.; data curation, Q.Z.; writing—original draft preparation, Q.Z.; writing—review and editing, Q.Z. and F.D.; visualization, H.D.; supervision, F.D.; project administration, F.D.; funding acquisition, W.Z. All authors have read and agreed to the published version of the manuscript.

**Funding:** This research was funded by the Program of National Natural Science Foundation of China (No. 52365029, 52065005, 51775115), Gansu Province Outstanding Youth Fund Project (No. 20JR10RA560).

**Data Availability Statement:** Data are contained within the article. The data presented in this study are available on request from the corresponding author.

**Acknowledgments:** The authors also sincerely appreciate the comments and suggestions for modifications made by the editors and anonymous referees.

**Conflicts of Interest:** The authors declare no conflict of interest.

## Nomenclature

| | |
|---|---|
| $\partial_s$ | wear constant |
| $l$ | relative sliding distance of the particles/mm |
| $a$ | radius of the contact area/mm |
| $F_n$ | reaction force on the particle/N |
| $\theta$ | angle of the center of the corresponding arc/(°) |
| $A_0$ | cross-sectional area of the spherical indentation/mm$^2$ |
| $\delta_n$ | normal overlap/mm |
| $R_p$ | particle radius/mm |
| $E^*$ | equivalent modulus of elasticity |
| $H_e$ | hardness of the particles/Pa |
| $\sigma_c$ | yield stress of the particle/Pa |

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
