# Peer review of "Effect of Operating Parameters on the Mulching Device Wear Behavior of a Ridging and Mulching Machine"

_lubricants, doi:10.3390/lubricants12010019_

Round 1

Reviewer 1 Report

Comments and Suggestions for Authors

- The title of the article should be corrected and a more suitable alternative should be written for it. Currently, the title of the article is a bit ambiguous.

- Authors should modify the abstract and add the most important results of the research in quantitative form.

- It is recommended that the authors add more references in the introduction section.

- Also, the manner and order of presenting the articles should be such that the reader's mind is moved towards the innovation of this article by reading the introduction section.

- How did the authors use formula 1 in this research?

- The explanations given about the numerical simulation are very few. Authors should include the solution method used, contact conditions, boundary conditions and type of element used in the paper.

- My important question about the numerical simulation is how the authors have verified the numerical simulations and the results obtained from them?

- The results of the numerical simulation are currently presented in the form of contours. Authors should present the results of numerical simulations in the form of graphs and tables and quantitatively.

Reviewer 2 Report

Comments and Suggestions for Authors

The reviewed manuscript requires supplementation. Below are my main comments:

The Authors are obliged to supplement the literature review with current items. The latest publication is from 2021, and this is the end of 2023.

The Authors should prepare a list of symbols and markings used in the content of the manuscript.

Table 1 and Table 4. Please add symbols for the listed parameters.

Table 2. Are the given values the result of a single measurement? If so, I suggest taking more measurements because a single measurement does not provide significant information.

Figure 8. How many tests were performed for the results shown in these three graphs?

Conclusions should be completed after making corrections in earlier parts of the manuscript.

Reviewer 3 Report

Comments and Suggestions for Authors

In general, reviews seek to improve a paper submitted for evaluation, but in this case I think the paper is well presented and documented. It is a finite element study, maybe know a little more about the mesh size determination, or if there was a sensitivity study. Otherwise it is an article that contributes in a niche of interest and that can be extended and applied to other similar cases (an idea of the extension of the scope could be given in future lines of the conclusions).

Reviewer 4 Report

Comments and Suggestions for Authors

The article studied the influence of operating parameters on mulching device wear behavior of ridging and mulching machine. I suggest accepting it after minor modifications.

1. Suggest changing the title to Effect of operating parameters on mulching device wear behavior of riding and mulching machine.

2. Please provide the affiliation information of all authors.

3. The first paragraph in the introduction has too many words. Please use academic language to briefly introduce the research background.

4. Does Figure 1 require a reference? Please indicate.

5. Should units be added to Formula 1? For example, l is the relative sliding distance of the particles, mm

6. Figure 3 is not very clear, it is recommended to increase the clarity or increase the local magnification of the image.

7. It is recommended to use the latest published references.

8. Is the use of the word Archard wear in Figures 7 and 10 accurate? Please verify.

9. The picture titles in Figures 8 and 9 should be more diverse. The vertical coordinates of Figures 8 and 9 should be expressed as the wear volume.

10. The study and addition of the following papers shall strengthen the weak literature review executed for this work:

https://doi.org/10.1016/j.jmrt.2023.04.036

Comments on the Quality of English Language

Minor editing of English language required

Round 2

Reviewer 1 Report

Comments and Suggestions for Authors

The manuscript can be accepted in the present form.

Reviewer 2 Report

Comments and Suggestions for Authors

The revised manuscript looks better than its previous version. 
I have no more comments.